# Retinal Vascular Complications in Cocaine Abuse: A Case Report and a Literature Review

**DOI:** 10.3390/jcm13247838

**Published:** 2024-12-22

**Authors:** Marta Armentano, Ludovico Alisi, Giacomo Visioli, Maria Carmela Saturno, Arianna Barba, Alessio Speranzini, Giuseppe Maria Albanese, Magda Gharbiya, Ludovico Iannetti

**Affiliations:** 1Department of Sense Organs, Medicine and Dentistry Faculty, Sapienza University of Rome, Viale del Policlinico 155, 00161 Rome, Italy; marta.armentano@uniroma1.it (M.A.); ludovico.alisi@uniroma1.it (L.A.); mariacarmela.saturno@uniroma1.it (M.C.S.); arianna.barba@uniroma1.it (A.B.); alessio.speranzini@uniroma1.it (A.S.); giuseppemaria.albanese@uniroma1.it (G.M.A.); magda.gharbiya@uniroma1.it (M.G.); 2Ophthalmology Unit, Head and Neck Department, Policlinico Umberto I University Hospital, 00161 Rome, Italy; ludovicoiannetti@gmail.com

**Keywords:** cocaine abuse, retinal vasculitis, retinal ischemia, retinal hemorrhage, vascular occlusion, vitrectomy, frosted branch angiitis, ocular complications, substance abuse, retinal pathology

## Abstract

This comprehensive review examines the ocular vascular complications of cocaine use, focusing on its effects on retinal vasculature and inflammation. A rare case of bilateral frosted branch angiitis (FBA) in a 48-year-old man with a history of intranasal cocaine abuse is presented as an illustrative example to stimulate discussion. The case highlights severe retinal ischemia and vascular sheathing, with no identifiable infectious or autoimmune cause, ultimately complicated by systemic vascular events. Integrating this case with a review of the literature, we discuss cocaine’s vasoconstrictive and immunomodulatory effects and their role in retinal pathology, including vasculitis, vascular occlusions, hemorrhages, and optic neuropathy. Although often a diagnosis of exclusion, and with rare and poorly understood consequences, this review underscores the importance of considering cocaine abuse in the differential diagnosis of complex retinal presentations.

## 1. Introduction

Retinal vascular complications from cocaine abuse are a considerable diagnostic challenge, often requiring the exclusion of more common causes and, in some cases, complicated by patients’ reluctance to disclose substance use. Derived from the *Erythroxylon coca* plant, cocaine has a complex history of both medical and recreational applications [1]. Initially believed to be safe, it was once widely incorporated into products like toothache remedies and energy tonics [2]. Today, cocaine exists in two main forms for recreational use: cocaine hydrochloride, a white powder typically snorted or injected; and crack cocaine, a free base form smoked in small, solid “rocks” [3]. The abuse of cocaine is a global health issue, and it has been estimated that over 20 million people are currently cocaine users [4].

Cocaine has well-known vasoconstriction activity linked to its vasoactive metabolite benzoylmethylecgonine, which enhances sympathetic activity, serving as a sodium channel blocker [5]. Cocaine also enhances the sympathetic and dopaminergic effect through the inhibition of monoamine reuptake, and it can augment the release of endothelin-1, a powerful vasoconstrictor agent [6,7]. The cardiovascular system is very susceptible to cocaine toxicity, causing irreversible structural and functional damage. In particular, cocaine has a strong sympathomimetic activity on vascular smooth muscle cells alfa-1 receptors and increases catecholamines at synaptic terminations, and the release of endothelin-1. This contributes to vasoconstriction, high blood pressure, and increased cardiac rate, with a subsequent increase in myocardial oxygen need. In the long run, this condition can lead to atherosclerosis, thrombotic clot formation, cardiac rhythm alterations, and ischemia with a high risk of myocardial infarction [8]. Vasoconstriction and high blood pressure can be responsible for an increase in vascular resistance and subsequent hypertrophic cardiomyopathy, thus favoring left ventricular dysfunction, low ejection fraction, and heart failure [9]. Other vascular districts strongly damaged by cocaine abuse are the kidneys and the brain. Hemodynamic alterations in the renal district can induce glomerular dysfunction, interstitial nephritis, and chronic microangiopathy, leading to malignant hypertension or infarction with organ failure [10]. Cocaine can also pass the blood–brain barrier. The serotonergic and dopaminergic boost in cocaine addiction can cause seizures and migraine, also showing a neurotoxic effect [11]. Vascular alterations in this district may cause ischemic or hemorrhagic stroke [12].

The eye, with its vascular structures, is affected by cocaine’s effects. In the literature, different vascular alterations have been identified. Retinal hemorrhages, retinal vein or artery occlusions, microvascular alterations, vascular caliber, and branch modifications were reported [13]. Several papers also indicate that cocaine’s effects are not limited to the retina. Trimarchi et al. documented sinonasal osteocartilaginous necrosis in cocaine abusers, which can indirectly affect the eye due to the anatomical proximity of the nasal and ocular structures [14]. Such findings suggest that cocaine’s impact on the body can have cascading effects, potentially leading to complications that involve multiple systems, including the eye.

In this context, this study documents the first known case of bilateral frosted branch angiitis in a patient with cocaine addiction. To set the background for this case, we first provide a review of the literature on cocaine-induced ocular vascular complications, consolidating current evidence on the drug’s impact on retinal vasculature. Aiming to clarify the association between cocaine use and ocular disorders, particularly retinal inflammation and vascular occlusions, we hope to raise awareness among ophthalmologists and other clinicians about this rare but noteworthy complication. Considering that the ophthalmologist could be the first healthcare provider to suspect cocaine abuse, early recognition of these retinal signs may enable timely intervention, potentially preventing more severe and irreversible systemic symptoms.

## 2. Retinal Involvement in Cocaine Abuse

The following sections, summarized in Table 1, explore cocaine-related ocular complications, particularly those impacting retinal and vascular structures. This discussion will draw on case reports and studies that highlight the range of retinal and optic nerve effects observed in patients with a history of cocaine use, providing insights into both common and rare manifestations documented in the literature.

### 2.1. Retinal Vasculitis and Inflammatory Manifestations

Retinal vasculitis is a group of diseases mediated by the activation of the immune system leading to vasoconstriction and endothelial dysfunction. Cocaine abuse, with its impact on vascular tissues, can also act as a trigger for the immune system in the retinal vessels.

This mechanism can lead to posterior uveitis. We previously discussed how cocaine abuse could have acted as an immunological trigger in conditions like Eales disease, a form of retinal vasculitis characterized by neovascularization [22]. A similar case has been described by Roohipourmoallai et al. in a patient affected by sarcoidosis who smoked crack cocaine for 8 years. The patient presented with a compromised visual acuity bilateral retinal vasculitis detected by fluorescein angiography with ischemic areas, neovascularizations, and vitreous hemorrhages. Moreover, the patient showed an epiretinal membrane in both eyes, with a lamellar macular hole in the right eye and a full-thickness macular hole in the left eye. The authors concluded that the vasculitis characteristics and vitreoretinal interface abnormalities observed in the patient were not typical of sarcoidosis. Therefore, they suggested that the posterior uveitis was not directly caused by sarcoidosis but was instead attributable either to cocaine abuse or to an autoimmune phenomenon triggered by cocaine use [21].

Another report described a case of pyoderma gangrenosum and Wegener granulomatosis-like syndrome induced by cocaine, indicating that cocaine abuse may mimic primary autoimmune diseases, which may also have ocular manifestations [32].

Cocaine use has also been implicated as a potential trigger for a case of neuromyelitis optica, characterized by positivity for anti-aquaporin 4 antibodies. Cocaine can pass the blood–retina barrier and hypothetically induce damage to the astrocytes, leading to autoimmune sensitization [33].

However, while these case reports on the association between cocaine use and autoimmune diseases provide valuable insights, it is essential to recognize that they represent isolated and rare occurrences. Consequently, a direct link between cocaine use and autoimmune ocular conditions cannot yet be established with certainty.

### 2.2. Vascular Occlusions

Retinal vascular occlusions are some of the most significant ocular complications linked to cocaine use. Several studies and case reports have documented various retinal vascular events associated with cocaine abuse, emphasizing the drug’s vasoconstrictive properties that can lead to acute ischemic events in the retina. Kannan et al. reported a case of cilioretinal artery occlusion following intranasal cocaine insufflation [15]. Similarly, Friedman et al. documented a central retinal artery occlusion and optic nerve head ischemic event in a patient with a history of cocaine abuse [16]. These cases highlight the potential for cocaine to precipitate severe arterial occlusions in the eye. Additionally, massive retinal arterial ischemia involving the posterior pole, with varying degrees of irreversible vision impairment, has been reported [34,35].

Cocaine use has also been associated with venous occlusions. Campbell et al. described a case of unilateral central retinal artery occlusion together with bilateral central retinal vein occlusion [36]. Similarly, Zhou and Jiang documented a case of branch retinal vein and artery occlusion following intranasal cocaine abuse [37].

Furthermore, Leung et al. observed early retinal vascular abnormalities in African American cocaine users, noting increased venular caliber associated with retinal hypoxia and systemic inflammatory markers [13]. This suggests that cocaine use may contribute to retinal vascular changes even before symptomatic occlusions occur.

The pathophysiological mechanisms underlying these vascular complications remain speculative, as they are not fully understood and may vary based on specific patient conditions [38]. Cocaine is known to induce a prothrombotic state, leading to an increased risk of thrombosis and subsequent ischemia in the retinal vasculature. Fortenbach and Modjtahedi highlighted that cocaine can cause ischemic injury through local vasoconstriction, hypertension, and vascular injury, which collectively contribute to retinal damage [17]. This is consistent with findings from Sharma and Ramirez-Florez, who noted that cocaine use is associated with various forms of retinal ischemia, including central and branch retinal artery occlusions [11].

In patients with vascular occlusions, additional retinal alterations associated with cocaine use have been described. Fortenbach and Modjtahedi reported bilateral enlargement of the foveal avascular zone on optical coherence tomography angiography in cocaine users, suggesting that cocaine can cause structural vascular changes in the retina, potentially compromising visual function [17]. This aligns with the work of Hulka et al., which indicated that cocaine use may impair color vision and cognitive function, potentially due to alterations in the retinal dopamine system [39]. The interplay between retinal health and cognitive function is particularly concerning, as it suggests that cocaine’s effects extend beyond mere vascular complications to involve neurochemical pathways critical for visual processing. The effects of cocaine on blood vessels are further supported by Bachi et al., who reviewed the mechanisms of cocaine-induced vasoconstriction and endothelial dysfunction, emphasizing the acute and chronic vascular effects of cocaine on various organ systems, including the eyes [5,37].

### 2.3. Hemorrhagic Events

One notable complication associated with intranasal cocaine abuse is Valsalva retinopathy, which has been documented in several case reports. Karasavvidou et al. described a case where intranasal cocaine use led to preretinal hemorrhage, highlighting the potential for significant visual impairment resulting from cocaine-induced vascular changes [18]. This aligns with findings from Pinilla et al., who reported a case of macular preretinal hemorrhage after intranasal cocaine consumption. The patient had to undergo a vitrectomy with posterior hyaloid removal to allow blood drainage [19]. These studies collectively suggest that cocaine’s effects on the catecholaminergic pathway determine an acute increase in blood pressure, leading to hemorrhages and acute ischemic events in the retina. A recent paper reported a case of Purtscher-like retinopathy in the context of atypical hemolytic uremic syndrome, allegedly triggered by cocaine use. The patient exhibited “Purtscher flecken”, multiple cotton wool spots, and intraretinal hemorrhages [40]. Interestingly, reports of cocaine-induced retinal hemorrhagic events are not limited to adults. Some data regarding intrauterine exposure to cocaine are also available. A study enrolling 30 women with a history of cocaine or multiple drug abuse during pregnancy revealed significant retinal anomalies in their newborns. Drug exposure can compromise placental blood supply, leading to fetal hypoxemia. At a retinal level, this condition can be responsible for vascular disruption, retinal ischemia, and hemorrhagic lesions [20].

A study has shown that prenatal exposure to cocaine can lead to significant retinal damage in newborns, including deep intraretinal hemorrhages indicative of chronic ischemia [11]. Similar persistent hemorrhagic lesions were reported in infants born from cocaine-abusing mothers by Silva-Araújo et al. [20]. However, it is important to note that retinal hemorrhages are a common clinical finding in newborns, making it challenging to distinguish between normal birth-related hemorrhages and those induced by substances like cocaine [41]. Interestingly, in murine prenatal models of cocaine exposure, alterations were observed in the structural organization of photoreceptors, including an increased number of displaced rods. These effects may result from the direct action of cocaine or from cocaine-induced ischemia/hypoxia [42]. In light of these reports, increased attention is advisable for infants born to drug-addicted mothers, not only due to a higher risk of infections but also because the presence of perinatal retinal hemorrhages or other retinal alterations should prompt ophthalmologists to consider the child’s social and family context [43].

### 2.4. Macular Involvement and Neovascularizations

Ascaso et al. reported a rare case of maculopathy, which presented with impaired color vision in a patient with a history of long-term intranasal cocaine use. Optical coherence tomography (OCT) scans were negative, while fluoresceine angiography revealed a pooling on the foveal area [23]. Arepalli et al. also reported a case of a 26-year-old patient who experienced monolateral acute visual loss after cocaine use. After the exclusion of cerebral vascular causes of sudden visual loss, the clinicians highlighted a paracentral acute middle maculopathy (PAMM). The OCT examination revealed a hyper-reflectivity of the inner retinal layer in the involved area [24]. PAMM is usually attributable to deep retinal capillary ischemia, typically presenting as hyper-reflectivity of the inner retina on OCT imaging. The most well-known causes of PAMM include systemic conditions such as hypertension, diabetes, and sickle cell disease, as well as acute events like retinal vascular occlusions and systemic hypotension. Cocaine abuse could also be considered a potential contributor to PAMM due to its prothrombotic and vasoconstrictive effects, which may induce retinal ischemia [44].

Another way cocaine can affect the retina is not only through its indirect impact on blood vessels but also via the deposition of tiny microcrystals from the substances injected intravenously. These microcrystals can obstruct small retinal vessels, leading to a condition known as talc maculopathy. At the fundoscopic examination, this condition can appear as multiple sparkling crystals deposited within the posterior pole. Different intravenous drugs are linked to this ocular complication, not exclusively cocaine. Fluorescein angiography is strongly recommended to highlight ischemic areas and neovascularization [45,46]. The talc emboli obstructing retinal and choroidal circulation are responsible for shunt formation, vessel tortuosity, and retinal hypoxia. These conditions can drive the development of peripheral or optic disc neovascularizations [25]. Interestingly, in an isolated case reported in the literature, a retinal cocaine crystal was observed to stimulate the development of a non-infectious foreign body granuloma. The patient presented with unilateral visual impairment, hemicentral scotoma, and intraretinal crystal deposits accompanied by exudates. The foreign body embolus appeared prominent and glistening, without associated inflammatory signs. Over time, the retinal lesion developed into a granuloma, surrounded by a fibro-retinal membrane and smaller crystals, which contributed to intraretinal edema and collateral vessel formation. Given the lack of inflammatory or infectious indicators, the clinicians adopted a conservative, observational approach. Similar to other areas of the body, the authors suggested that the accumulation of talc or other diluents might lead to the formation of an intraretinal foreign body granuloma. However, as this is an older case report, the absence of modern retinal imaging technology limits the potential for further speculation [47].

Another complication associated with cocaine consumption is acute macular neuroretinopathy (AMN). AMN is characterized by decreased vision, paracentral scotoma, and distinct wedge-shaped hyper-reflective lesions in the outer plexiform layer, which are detectable with OCT. These findings are typically accompanied by generalized retinal thinning and disruption of the ellipsoid zone. The underlying cause is thought to be ischemia of the deep capillary plexus, resulting in macular hypoperfusion [48,49]. Fluorescein angiography and indocyanine green angiography can help visualize the hypoperfusion state, revealing choroidal filling defects, dilation of perimacular capillaries, and punctate early hypofluorescence with late staining. However, it is often challenging to distinguish cocaine abuse from other systemic conditions that may contribute to AMN, such as cardiovascular shock, severe hypotension, trauma, eclampsia, and contraceptive use [50]. Therefore, cocaine should be considered a potential cofactor in AMN, with diagnosis ideally involving the exclusion of these other conditions.

### 2.5. Optic Neuropathy and Retinal Nerve Fiber Alterations

Gemelli et al. conducted a study with the aim of identifying alterations in the retinal nerve fiber layer (RNFL) of cocaine users compared to age-matched healthy controls. The results showed a reduced RNFL thickness in chronic drug users, due to microischemic damage. This process is also known to happen in other hematologic, metabolic, and inflammatory pathologies determining ischemia, such as diabetic retinopathy or hemoglobinopathies [26]. Moreover, damage to the retinal ganglion cell axons is closely associated with neurodegeneration, representing a hallmark of this process. Cocaine users are exposed to a high risk of premature cerebral atrophy and neurodegeneration [51,52,53]. Cocaine-induced damage to the optic nerve has also been reported. Goldberg et al. described three cases of optic neuropathy following intranasal cocaine use. All three patients suffered from severe forms of osteolytic sinusitis. The authors hypothesized two pathogenetic mechanisms. On the one hand, in all three cases, the inflammatory process had spread from the paranasal sinuses into the orbital cavity, involving the orbital apex; on the other hand, a vascular ischemic mechanism is also conceivable [27]. Cocaine can also trigger the development of toxic ischemic optic neuropathy. This pathology usually presents with different degrees of vision loss, dyschromatopsia, peripapillary hemorrhages, and central scotoma. Cocaine can determine irreversible damage to the papillomacular bundle consistent with bitemporal RNFL loss [54]. In line with these findings, Newman et al. described a case of optic neuropathy in the context of osteolytic sinusitis in a patient who reported daily intranasal cocaine use. In this patient, resonance imaging documented diffuse disruption of the nasal cavity, paranasal sinus, and the floor of the anterior cranial fossa was associated with color vision impairment, diminished visual acuity, and a relative afferent pupillary defect in one eye. The optic nerve presented temporal pallor with modest swelling, which also involved RNFL fibers and caused a perimetric centrocecal scotoma. Long-term intranasal cocaine use is considered responsible for ischemic damage, necrosis, and perforation of nasal and paranasal bones and cartilage. The optic nerve and the surrounding orbital structures can be involved in the inflammatory process and directly suffer from cocaine’s vasoconstrictive effects [28]. Particular cases of osteolytic sinusitis following intranasal cocaine consumption have been described by Lin et al. and Shen et al. Both patients developed chronic orbital inflammation as an extension of the inflammatory foci from the disrupted sino-nasal bones. Both patients presented with proptosis, strabismus, diplopia, abnormal extraocular motility, periorbital edema, conjunctival hyperemia, and orbital pain. Recurrent episodes of orbital inflammation and infection led to posterior ischemic optic neuropathy on a compressive and inflammatory base, with irreversible vision loss [55,56]. These findings align with those reported by Simerink et al. The authors described two cases of optic neuropathy following the expansion of a granulomatous and fibro-inflammatory reaction to the orbital tissues, during cocaine abuse. In this condition, the optic nerve can undergo an ischemic injury due to the compression and infiltration determined by the rapidly growing inflammatory lesion involving the orbital apex [57].

Finally, an exceptional case of Leber’s hereditary optic neuropathy (LHON) linked to cocaine abuse has been flagged. LHON is caused by a maternal-inherited mitochondrial DNA mutation, with complete expressivity and variable penetrance. It has been suggested that the etiological mechanism underlying this disease may be influenced not only by genetic factors but also by environmental ones. Cocaine is known to trigger the production of reactive oxygen species, which can cause mitochondrial damage and reduce mitochondrial functional reserve. It is therefore plausible to hypothesize that, in a patient with a specific mitochondrial DNA mutation associated with LHON, cocaine use played a decisive role in triggering the onset of clinical symptoms. This impact on the delicate mitochondrial functional balance likely contributed to mitochondrial failure [58].

### 2.6. Photoreceptor Dysfunction and Electrophysiological Abnormalities

Electroretinographic (ERG) abnormalities have been observed in cocaine-addicted patients, particularly during the withdrawal phase. Roy et al. studied ERG patterns in 20 patients with cocaine addiction in withdrawal, comparing their results with those of healthy controls matched for age, ethnicity, and sex. They found a significant reduction in blue cone activity in the cocaine-withdrawn group, attributing this effect to the well-documented dopamine dysfunction caused by cocaine addiction. Dopamine, a crucial neurotransmitter at the retinal level, is produced by dopaminergic amacrine and interplexiform cells, which mediate interactions between photoreceptors and bipolar neurons [29]. Cocaine withdrawal can reduce dopamine availability, disrupting retinal processing and cone function. This dysfunction is associated with blue–yellow color vision impairment [30] and correlates with cocaine craving [59]. The authors also proposed that ERG responses might help identify ex-cocaine users at higher risk of relapse [31]. As discussed above, it is challenging to isolate cocaine as the sole etiologic factor in such observational studies. Cocaine-addicted patients often face additional social and health-related conditions—such as poor nutrition, chronic alcoholism, infectious diseases, and other comorbidities—that can independently contribute to retinal and optic pathway damage, measurable through electrophysiology [60]. These complex factors must be taken into account when interpreting findings related to retinal damage and visual processing impairments in this population.

## 3. A Case Report of Bilateral Frosted Branch Angiitis in Cocaine Abuse

A 48-year-old Caucasian man presented to our emergency room reporting a history of progressive vision loss for one year in both eyes followed by a sharp visual deterioration, which mainly affected the left eye (LE).

At the presentation, the patient reported a 5-year history of cocaine abuse via the intranasal route. The patient suffered from chronic renal disease stage IV with three-times-a-week dialytic therapy while waiting for a kidney transplant and systemic hypertension treated with doxazosin. He underwent biliointestinal bypass surgery 20 years ago because he was obese. He did not report recent respiratory, genitourinary, or gastrointestinal infections.

The patient presented with best-corrected visual acuity (BCVA) of light perception in both eyes (BEs). On slit lamp examination, neither eye showed signs of iris neovascularization or cells and flare in the anterior chamber. The intraocular pressure was 12 mmHg in BEs. The fundus examination of BEs showed vitreous hemorrhages (VHs), greater in the LE, with fibrovascular tractive membranes at the posterior pole, a pale optic disc, and frosted branch angiitis (FBA) with diffuse retinal ischemia (Figure 1). As Appendix A, we include a video showing a few seconds of the pars plana vitrectomy (PPV), illustrating the accumulation and movement of the vitreous hemorrhages above the optic disc as well as the tractive membranes (Appendix A).

Macular spectral OCT in BEs confirmed the presence of vitreous hemorrhage and tractional membranes over the macular region (Figure 2). OCT also showed an overall reduction in retinal thickness, with hyper-reflectivity of inner layers. Fluorescein angiography was not performed due to a previous severe allergic reaction to fluorescein.

Given the clinical presentation, several laboratory tests were performed to rule out the main causes of FBA. Serology for herpes simplex virus (HSV) 1 and 2, cytomegalovirus (CMV), varicella zoster virus (VZV), and Epstein–Barr virus (EBV) was negative for recent infections. Additionally, tests for autoantibodies, including antinuclear antibodies, extractable nuclear antigen antibodies, antineutrophil cytoplasmic antibodies, anti-mitochondrial antibodies, anti-cardiolipin antibodies, lupus anticoagulant, and anti-beta-2-glycoprotein-I antibodies, were negative. Peripheral blood T- and B-cell subpopulations were within normal ranges.

A 25G pars plana vitrectomy (PPV) was performed in the LE to treat VHs and remove tractional membranes. Peripheral laser photocoagulation of the ischemic areas was conducted. Aqueous and vitreous taps were obtained during the procedure to perform further laboratory tests. Polymerase chain reactions (PCRs) for HSV1, HSV2, VZV, EBV, CMV, and Toxoplasma gondii were performed for both samples and the results were negative. The day after surgery, the postoperative course showed superficial punctate keratitis in the inferior corneal sectors, and a slight reaction in the anterior chamber (flare +/− and cells 1+). The fundus examination revealed vitreous haze (4+), and the posterior pole and peripheral retina could not be visualized. The patient was given therapy with steroids, antibiotics, and FANS drops (betamethasone and chloramphenicol six times daily and bromfenac two times daily). One week after surgery, superficial punctate keratitis was observed in all corneal sectors, the anterior chamber was deep with cells +/− but the posterior pole and peripheral retina were still not visible due to vitreous haze 4+. Preservative-free artificial tears every 2 h were added to the previous therapy. Two weeks after surgery, the BCVA was hand motion. Superficial punctate keratitis was not improved but the anterior chamber showed no reaction, and vitreous haze was reduced (1+). The fundus examination showed resolution of VHs, with good visualization of the pale optic disc and frosted branch angiitis and slight pigmentation of the peripheral photocoagulation areas. The patient was advised to discontinue antibiotic drops and to taper down steroid drops over four weeks. One week after the last examination, the patient showed up reporting tearing and mucous discharge in the LE. A slit lamp examination showed an inferior paracentral corneal ulcer with elevated gray–white edges and positive fluorescein staining, which was diagnosed as a neurotrophic ulcer (Figure 3).

Central corneal esthesiometry was less than 2 mm and swabs for bacteria or fungi were negative. Preservative-free artificial tears were given every hour and netilmicin drops two times daily. After three days, the clinical examination showed no improvement in the corneal ulcer, so a therapeutic contact lens was applied. The last examination was two months after surgery. BCVA was 1.5 LogMar in the LE. A slit lamp examination showed the contact lens was well positioned, with a reduction in the corneal ulcer diameter, absence of an anterior chamber, and vitreous reaction. Fundus examination showed no significant differences. The patient was given another appointment after two weeks to check for any improvement in the corneal ulcer and to plan PPV in the RE but one week after our last examination the patient died from complications related to the systemic vascular diseases.

## 4. Discussion

FBA is a sporadic condition, reported in a couple of hundred cases. The FBA pathogenesis is mostly still unknown. The typical development of FBA after a multifactorial prodromal illness suggests that a hypersensitivity reaction to various infectious agents may trigger FBA through a common pathway, possibly involving the deposition of immune complexes [61,62]. Kleiner historically recognized three forms of FBA: type 1, mediated by direct invasion of cancerous cells; type 2, usually secondary to viral infections; and type 3 or idiopathic [63]. We can only hypothesize the causative effects of cocaine in the development of FBA. It is well known that cocaine can induce vasculitis with clinical manifestations close to primary idiopathic antineutrophil cytoplasmic autoantibody (ANCA) vasculitis. The mechanism is still poorly understood but may be related to vascular ischemia leading to the formation of neutrophil extracellular traps, with the final step being the production of ANCA [64]. The involvement of retinal tissue in this specific inflammation pathway ranges considerably from anterior uveitis to scleritis and severe vaso-occlusive disease [65,66]. Notably, our patient was negative for the whole antibody panel but, as mentioned above, due to the asynchronous pattern of expression, the involvement of autoimmunity cannot be ruled out for certain [65]. Moreover, the probable addition of Lemvisole, a contaminant of around 70% of cocaine globally, potentially contributed to the worsening of vasculitis [67]. Levamisole-adulterated cocaine (LAC) vasculopathy is a devastating syndrome, usually limited to the skin, characterized by thrombotic vasculopathy with a deposition of mixed immunocomplex and C3 in the vessel walls [68,69]. Similarly to cocaine-induced vasculopathy, LAC is often associated with ANCA positivity [70]. Lastly, we can hypothesize that talc retinopathy may have contributed to the development of the clinical presentation, even though our patient reported no clinically detectable intraretinal talc deposits. This condition has been related to vascular occlusions, ischemia, and new vessel development [45,71]. The amplitude of the ocular involvement is also represented by the subsequent development of NK following vitreoretinal surgery. NK is a rare condition characterized by a loss in corneal sensitivity [72,73]. The development of NK is a known side effect of cocaine snorting. It has been suggested that the direct toxicity of cocaine on corneal nerves may be responsible for the development of such conditions [74,75]. A study indicates that even ten years after discontinuing cocaine use, corneal nerves may still exhibit abnormalities [76]. Overall, it would be recommended to check for corneal sensitivity in all cocaine users, even without evident corneal defects. Interestingly, Gabriele et al. reported in a case report a similar presentation to our case in an intravenous cocaine user. In this case, the patient presented with FBA and optic nerve hyperemia, and further analysis of the vitreous revealed a Fusarium dimerum endophthalmitis [77]. Unfortunately, we did not perform microbiological analysis on the vitreous so we cannot rule out an atypical infection. However, our patient did not show any clinical signs of potential infective involvement of the vitreous cavity. We report a very unusual case of FBA with severe retinal and corneal involvement in a cocaine user with a full negative infectious and autoimmune panel. The case reported may represent a secondary FBA resulting from the combined effects of cocaine abuse, chronic renal and vascular conditions, and potentially undiagnosed autoimmune factors. The interaction of these chronic conditions likely contributed to the development of FBA, highlighting the importance of considering multifactorial etiologies. However, it is important to note that the simultaneous presence of cocaine abuse alongside an extremely rare retinal finding does not necessarily imply a causal relationship [78]. Therefore, it is essential to increase the scientific evidence to draw general conclusions about a direct relationship.

## 5. Conclusions

We highlighted the diverse retinal manifestations of cocaine abuse and reported the first case of FBA with hemorrhagic involvement and fibrovascular proliferation in a cocaine user. The negativity of all serological tests, antibodies, and viral infections suggests a potential causative role of cocaine, given its extensive effects on both autoimmune processes and vascular tissues. However, in diagnoses of exclusion, it is impossible to establish absolute certainty regarding causation, as only the contemporaneity of phenomena can be confirmed. In complex retinal cases, it is important to evaluate all possible underlying causes, including substance abuse, which patients may not readily disclose during initial history-taking, to avoid delays in diagnosis and timely intervention.

## Figures and Tables

**Figure 1 jcm-13-07838-f001:**
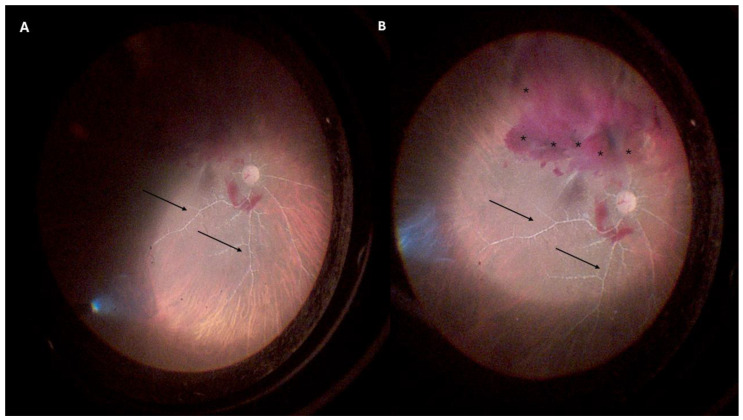
Intraoperative images of the left eye retina. (**A**) The image highlights vascular sheathing typical of frosted branch angiitis (indicated by black arrows) and a pale optic disc. (**B**) A broader view shows vitreal hemorrhages (black asterisks), tractive macular membranes, optic disc pallor, and prominent vascular sheathing (black arrows).

**Figure 2 jcm-13-07838-f002:**
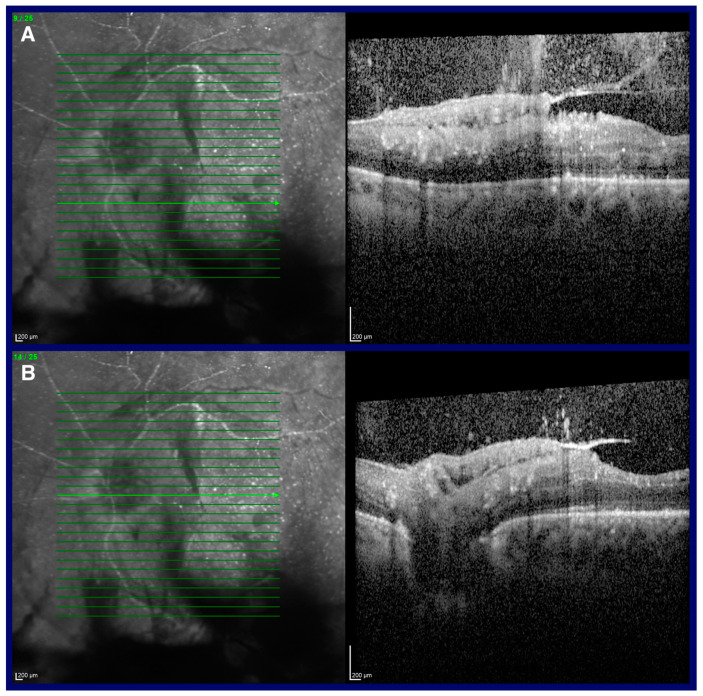
Optical coherence tomography (OCT) scans of the left eye. The upper panels (**A**) show a dense fibrovascular tractive membrane at the posterior pole, causing significant distortion of the underlying retinal layers. The lower panels (**B**) present another section of the same scan at the level of the optic disc, demonstrating that the tractive membrane involves the entire posterior pole.

**Figure 3 jcm-13-07838-f003:**
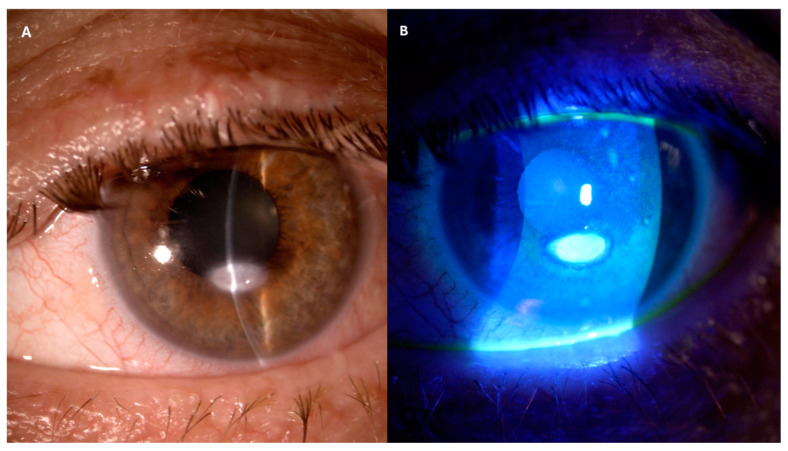
Slit lamp images showing paracentral corneal ulcer: white light (**A**); fluorescine coloring (**B**).

**Table 1 jcm-13-07838-t001:** Key features of retinal manifestations and involvement in cocaine abuse.

Event	Available Evidence	Key References
Vascular Occlusion	Cocaine’s vasoconstrictive properties may lead to retinal vascular occlusion, including cilioretinal artery occlusion and bilateral vascular occlusions, resulting in ischemic events and irreversible vision impairment.	Kannan et al., 2011 [15]; Friedman et al., 2010 [16]; Fortenbach & Modjtahedi, 2020 [17]
Hemorrhagic Events	Valsalva retinopathy, macular preretinal hemorrhage, and hemorrhagic lesions in newborns with intrauterine cocaine exposure highlight the hemorrhagic risks.	Karasavvidou et al., 2019 [18]; Pinilla I, 2007 [19]; Silva-Araújo A, 1996 [20]
Retinal Vasculitis/Inflammatory Manifestations	Cocaine can trigger retinal vasculitis through prothrombotic states and immune activation. Cases include exacerbated retinal vasculitis in conditions like Eales disease and sarcoidosis.	Roohipourmoallai et al., 2021 [21]; Iannetti et al., 2023 [22]; Bachi et al., 2017 [5]
Macular Involvement and Neovascularizations	Cocaine-induced maculopathy and paracentral acute middle maculopathy (PAMM), as well as talc maculopathy and neovascularization due to vascular obstructions from drug particles.	Ascaso et al., 2009 [23]; Arepalli et al., 2023 [24]; Tran KH, 2007 [25]
Optic Neuropathy and Retinal Nerve Fiber Alterations	Cocaine use is linked to optic neuropathy and RNFL damage, potentially exacerbated by sinusitis-related inflammation or neurodegenerative effects leading to visual field defects.	Gemelli H, 2019 [26]; Goldberg R, 1989 [27]; Newman NM, 1988 [28]
Photoreceptor Dysfunction and Electrophysiological Abnormalities	Electroretinographic studies show reduced blue cone activity during withdrawal, potentially due to dopamine dysregulation. Links to color vision impairment and risk of relapse have been noted.	Roy M, 1997 [29]; Desai P, 1997 [30]; Smelson DA, 1998 [31]

## Data Availability

Not applicable.

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
