# Peer review of "Retinal Vascular Complications in Cocaine Abuse: A Case Report and a Literature Review"

_jcm, 2024, doi:10.3390/jcm13247838_

Round 1
Reviewer 1 Report
Comments and Suggestions for Authors
I am happy to review this interesting manuscript on retinal vascular complications in cocaine abuse. Overall, the contents and information provided in this review is satisfactory and present enough details for the interest of the readers. However, some minor corrections need to be done based on the comments suggested.
Figures 1 and 2 presented within the manuscript needs to be clearer and properly labeled. They don't give much information.
Table 1 mentioned has no references added in the available evidence.
References are missing at multiple places such as Line 25-30, Line 91 etc. Please check throughout for the references.
Sentences needs to be reframed such as in line 104 "with anti-aquaporin 4 antibodies positivity". Line 346-349 "Also, autoantibodies (antinuclear antibodies, extractable nuclear antigen antibodies, anti-neutrophil cytoplasmic antibodies, anti-mitochondrial antibodies, anti-cardiolipin antibodies, lupus anticoagulant, anti-beta-2-glycoprotein-I antibodies) were negative. Peripheral blood T and B subpopulations were normal."
Line 321-322: Both eyes repeated
Comments on the Quality of English Language
Proofreading of the English language to be done, grammatical errors need to be checked throughout the manuscript.
Author Response
I am happy to review this interesting manuscript on retinal vascular complications in cocaine abuse. Overall, the contents and information provided in this review is satisfactory and present enough details for the interest of the readers.
Reply: We sincerely thank the reviewer for their valuable comments. We have also corrected some grammatical and structural English errors among the entire manuscript.
However, some minor corrections need to be done based on the comments suggested.
Figures 1 and 2 presented within the manuscript needs to be clearer and properly labeled. They don't give much information.
Reply: Thank you. We have improved the quality of all three figures and revised the captions for Figures 1 and 2 to enhance clarity and provide more detailed information. Let us know if any additional adjustments are needed.
Table 1 mentioned has no references added in the available evidence.
Reply: Thank you for your suggestion. We've added the references as required.
References are missing at multiple places such as Line 25-30, Line 91 etc. Please check throughout for the references.
Reply: We looked through the document and added the references where appropriate.
Sentences needs to be reframed such as in line 104 "with anti-aquaporin 4 antibodies positivity". Line 346-349 "Also, autoantibodies (antinuclear antibodies, extractable nuclear antigen antibodies, anti-neutrophil cytoplasmic antibodies, anti-mitochondrial antibodies, anti-cardiolipin antibodies, lupus anticoagulant, anti-beta-2-glycoprotein-I antibodies) were negative. Peripheral blood T and B subpopulations were normal."
Reply: Thank you. We have rephrased the abovementioned sentences.
Line 321-322: Both eyes repeated.
Reply: Amended.
Reviewer 2 Report
Comments and Suggestions for Authors
In the present manuscript, Armentano and colleagues review ocular complications associated with cocaine abuse. The authors also bring a case report to highlight the main findings regarding ocular complications in cocaine abuse, and finally the authors integrate the literature review and clinical features. That said, I would like to bring the author’s attention to the following:
1. Proulx & Tousignant (2020) have shown an extended review of drugs of abuse and ocular defects using different databases and papers published from 1980 to 2019. Individual substances such as cocaine are also included in the review and main findings are corneal damage, bilateral acute macular neuroretinopathy, maculopathy and thinning of retinal nerve. Larranaga-Cores et al. (2023) has reported Purtscher-like retinopathy after cocaine abuse. Overall, these cited key references are not listed in the review. Moreover, the authors should include the list of references for each clinical evidence listed on the table (Table 1).
2. As the authors have added a case report to exemplify clinical evidence, please check journal guidelines for inclusion of patient data and if ethics committee approval protocol should be included in a separate session.
3. Lines #173-175: prenatal exposure to cocaine could lead to retinal damage due to cocaine-induced ischemia/hypoxia reported in mice models to alter retinal layering patterns (Baptista & Ambrosio, 2024).
4. The authors present a case report of bilateral FBA in a 48-year-old patient with a history of cocaine abuse and other potential causes were discarded such as several viral infections and lupus. The patient had also chronic renal disease, systemic hypertension and critical clinical conditions that later evolved to death. The authors conclude that autoimmune diseases cannot be ruled out (lines 401-403). Is it possible that the case reported here is a secondary FBA case, resulting from a conjunction of cocaine abuse and other known (renal and vascular) and/or unknown (possible autoimmune disease) chronic circumstances?
Author Response
In the present manuscript, Armentano and colleagues review ocular complications associated with cocaine abuse. The authors also bring a case report to highlight the main findings regarding ocular complications in cocaine abuse, and finally the authors integrate the literature review and clinical features.
Reply: We are grateful to the reviewer for the insightful comments.
That said, I would like to bring the author’s attention to the following:
1) Proulx & Tousignant (2020) have shown an extended review of drugs of abuse and ocular defects using different databases and papers published from 1980 to 2019. Individual substances such as cocaine are also included in the review and main findings are corneal damage, bilateral acute macular neuroretinopathy, maculopathy and thinning of retinal nerve. Larranaga-Cores et al. (2023) has reported Purtscher-like retinopathy after cocaine abuse. Overall, these cited key references are not listed in the review.
Reply: thank you for suggesting this key references. We've added them as you suggested.
Moreover, the authors should include the list of references for each clinical evidence listed on the table (Table 1).
Reply: Thank you. Actually they were removed by mistake before the submission. Now the Table 1 provides the relevant references.
2) As the authors have added a case report to exemplify clinical evidence, please check journal guidelines for inclusion of patient data and if ethics committee approval protocol should be included in a separate session.
Reply: Thank you for raising this ethical concern. Informed consent for the publication of the patient’s data was obtained at the time of the patient’s admission to our hospital. We have now included this information in the appropriate section.
3) Lines #173-175: prenatal exposure to cocaine could lead to retinal damage due to cocaine-induced ischemia/hypoxia reported in mice models to alter retinal layering patterns (Baptista & Ambrosio, 2024).
Reply: Thank you for your suggestion. We added the suggested reference on murine models.
4) The authors present a case report of bilateral FBA in a 48-year-old patient with a history of cocaine abuse and other potential causes were discarded such as several viral infections and lupus. The patient had also chronic renal disease, systemic hypertension and critical clinical conditions that later evolved to death. The authors conclude that autoimmune diseases cannot be ruled out (lines 401-403). Is it possible that the case reported here is a secondary FBA case, resulting from a conjunction of cocaine abuse and other known (renal and vascular) and/or unknown (possible autoimmune disease) chronic circumstances?
Reply: We agree that the case reported could indeed represent a secondary FBA resulting from a combination of cocaine abuse, chronic renal and vascular conditions, and potentially undiagnosed autoimmune factors. The interplay between these chronic circumstances may have contributed to the development of FBA, and we acknowledge the importance of considering such multifactorial etiologies. We revised the manuscript to reflect this perspective more clearly. In the conclusions we state: "However, in diagnoses of exclusion, it is impossible to establish absolute certainty regarding causation, as only the contemporaneity of phenomena can be confirmed".